# The Multiple Effects of Vitamin D against Chronic Diseases: From Reduction of Lipid Peroxidation to Updated Evidence from Clinical Studies

**DOI:** 10.3390/antiox11061090

**Published:** 2022-05-30

**Authors:** Massimiliano Berretta, Vincenzo Quagliariello, Alessia Bignucolo, Sergio Facchini, Nicola Maurea, Raffaele Di Francia, Francesco Fiorica, Saman Sharifi, Silvia Bressan, Sara N. Richter, Valentina Camozzi, Luca Rinaldi, Carla Scaroni, Monica Montopoli

**Affiliations:** 1Department of Clinical and Experimental Medicine, University of Messina, 98100 Messina, Italy; 2Division of Cardiology, Istituto Nazionale Tumori-IRCCS-Fondazione G. Pascale, 80121 Naples, Italy; quagliariello.enzo@gmail.com (V.Q.); n.maurea@istitutotumori.na.it (N.M.); 3Experimental and Clinical Pharmacology, Centro di Riferimento Oncologico di Aviano (CRO) IRCCS, Via Franco Gallini 2, 33081 Aviano, Italy; alexfumi82@gmail.com; 4Oncology Operative Unit, Santa Maria delle Grazie Hospital, 80078 Naples, Italy; sergio.facchini@studenti.unicampania.it; 5Gruppo Oncologico Ricercatori Italiani, GORI Onlus, 33170 Pordenone, Italy; rdifrancia@iapharmagen.org; 6Italian Association of Pharmacogenomics and Molecular Diagnostics (IAPharmagen), 60126 Ancona, Italy; 7Department of Radiation Oncology and Nuclear Medicine, AULSS 9 Scaligera, 37100 Verona, Italy; francesco.fiorica@aulss9.veneto.it; 8Department of Pharmaceutical and Pharmacological Sciences, University of Padova, 35122 Padova, Italy; emailforsaman@gmail.com (S.S.); silvia.bressan.6@phd.unipd.it (S.B.); monica.montopoli@unipd.it (M.M.); 9Veneto Institute of Molecular Medicine, 35129 Padova, Italy; 10Department of Molecular Medicine, University of Padova, Via A. Gabelli 63, 35121 Padova, Italy; sara.richter@unipd.it (S.N.R.); carla.scaroni@unipd.it (C.S.); 11Endocrinology Unit, Department of Medicine (DIMED), University of Padua, 35100 Padua, Italy; valentina.camozzi@unipd.it; 12Department of Advanced Medical and Surgery Sciences, Internal Medicine COVID Center, University of Campania Luigi Vanvitelli, 81100 Naples, Italy; lucarinaldi@hotmail.it

**Keywords:** vitamin D, calcium homeostasis, cancer, immune system, infectious disease

## Abstract

Background: Vitamin D exerts multiple beneficial effects in humans, including neuronal, immune, and bone homeostasis and the regulation of cardiovascular functions. Recent studies correlate vitamin D with cancer cell growth and survival, but meta-analyses on this topic are often not consistent. Methods: A systematic search of the PubMed database and the Clinical Trial Register was performed to identify all potentially relevant English-language scientific papers containing original research articles on the effects of vitamin D on human health. Results: In this review, we analyzed the antioxidant and anti-inflammatory effects of vitamin D against acute and chronic diseases, focusing particularly on cancer, immune-related diseases, cardiomyophaties (including heart failure, cardiac arrhythmias, and atherosclerosis) and infectious diseases. Conclusions: Vitamin D significantly reduces the pro-oxidant systemic and tissue biomarkers involved in the development, progression, and recurrence of chronic cardiometabolic disease and cancer. The overall picture of this review provides the basis for new randomized controlled trials of oral vitamin D supplementation in patients with cancer and infectious, neurodegenerative, and cardiovascular diseases aimed at reducing risk factors for disease recurrence and improving quality of life.

## 1. Introduction

Vitamin D (Vit D) is a fat-soluble vitamin represented by two related fat-soluble substances, cholecalciferol (Vit D3) and ergocalciferol (Vit D2), both of which can be used to cure or prevent rickets [1]. These molecules are produced from 7-dehydrocholesterol, known as Pro-Vit D, which is activated to Vit D by ultraviolet light, usually in the dermis or epidermis [2]. These sterols are then transported to the liver, where they undergo 25-hydroxylation (to 25-OH Vit D), and then to the kidneys, where they undergo a second hydroxylation to the fully active molecules: 1,25-dihydroxycholecalciferol (calcitriol) and 1,25-dihydroxyergocalciferol [3]. Humans synthesize sufficient amounts of Vit D when they have adequate exposure to sunlight. Moreover, Vit D acts more as a hormone than a vitamin, binding to specific cytosolic receptors, mainly located in intestinal epithelial cells and osteocytes, as well as in numerous other tissues, including hematopoietic cells, hair follicles, adipose tissue, muscle, and the brain [4]. After binding to its cytosolic receptors, 1,25(OH)_2_D is transported to the nucleus, where the vitamin-receptor complex interacts with DNA and modulates gene expression to increase calcium uptake. The most important effects of Vit D are calcium absorption and bone resorption; however, it exerts many other effects, not all of whose clinical implications are yet fully known [5,6]. Vit D is involved in many physiological processes and plays a therapeutic role in human diseases, such as cancer, infectious diseases, osteoarticular, and cardiovascular diseases [7]. There are many conflicting data on the role of Vit D in both prevention and therapeutic effects in many diseases, especially heart disease and cancer. In general, conventional doses of Vit D are well tolerated, with no significant adverse effects [8]. High doses of Vit D can be toxic and lead to a number of signs and symptoms, but not to liver damage or jaundice. Unfortunately, the lack of rigorous clinical studies makes it difficult to draw clear conclusions about the therapeutic aspect of Vit D, especially in cancer patients [9].

## 2. Materials and Methods

A systematic search of the Medline and EMBASE databases was performed to identify all potentially relevant English-language scientific papers containing original research articles on the effect of Vit D on human health. Full texts written entirely in English with available abstracts and at least one of the following characteristics were considered: clinical and/or preclinical studies on the role of Vit D in immune system homeostasis; cancer; infectious and cardiovascular diseases; and pharmacological processes, including pharmacokinetic aspects and risk of interactions. Boolean operators AND/OR were used to combine search terms. The following search strings were used in PubMed: “Vitamin D OR Vit D AND cancer” OR “Vitamin D OR Vit D AND infection” OR “Vitamin D OR Vit D AND cardiovascular” OR “Vitamin D OR Vit D AND immune system”. Simultaneous research was performed on the Clinica Trial Register. The databases were last accessed on 31 January 2022. MB, VQ, MM, and CS searched articles published in English until February 2022 and selected them on the basis of inclusion and exclusion criteria. A Preferred Reporting Items for Systematic Reviews and Meta-analyses (PRISMA) flow diagram [10], accessed on 20 March 2022, was created to summarize the systematic review process. Reports of the systematic review were performed according to the PRISMA guidelines and are shown in Figure 1 [11].

## 3. Endogenous and Exogenous Sources of Vitamin D

There are several sources from which Vit D can be obtained, such as food, supplements, or the skin (photosynthesis), as shown in Figure 2. Apart from fortified foods and fish (especially fatty fish), Vit D content in foods is generally not very high [12]. Vit D occurs in various forms known mainly as Vit D3 (cholecalciferol) and Vit D2 (ergocalciferol). Vit D3 is synthesized in the skin by exposure to ultraviolet (UV) radiation from the sun on its precursor, 7-dehydrocholesterol (Figure 2) [13,14]. Vit D2 (ergocalciferol) is another form of Vit D that is often used for fortification. In fungi (e.g., mushrooms), Vit D2 can also be produced by UVB irradiation of ergosterol, its precursor (Figure 2). Dietary sources of Vit D include fish, especially in the form of oil (Vit D3), egg yolk (Vit D3), and mushrooms (Vit D2), as well as artificially fortified cereal products and dairy products, such as milk (Vit D2 or D3) [15]. As mentioned above, the production of Vit D3 in the skin is independent of the involvement of enzymes and occurs via 7-dehydrocholesterol (7-DHC). In short, UV light in the spectral range of 290–320 nm UVB (under sunlight) can lead to the formation of pre-vitamin D3, which is isomerized to Vit D3 in a thermosensitive but non-catalytic process [16]. The rate of D3 formation depends on both UVB intensity and the degree of skin pigmentation [13]. This process can be affected by skin melanin, clothing, and sunscreens, which can prevent UVB from reaching the 7-DHC and, thus, negatively affect Vit D3 production.
Figure 2There are several sources of vitamin D: enriched food, supplements, and fortified foods. D2 and D3 are mainly a double bond between C22 and C23 and a methyl group at C24 in the side chain (Figure 3), so Vit D2 can be considered as the first Vit D analogue [17]. The structural differences to Vit D3 in the side chain are responsible for decreasing the affinity of Vit D2 to its target, the Vit-D-binding protein (DBP);therefore, it can be removed from the bloodstream more quickly [18]. This may limit its conversion to 25-hydroxyvitamin D (25OHD) (Figure 2) by at least some of the 25-hydroxylases yet to be described and alter its degradation by 24-hydroxyase (CYP24A1) [18]. Therefore, to achieve high blood levels of 25OHD compared to Vit D3 levels, D2 supplementation must be administered daily [19,20]. Nevertheless, 1,25(OH)_2_D2 and 1,25(OH)_2_D3 have similar affinity with the Vit D receptor (VDR) [17]. Regardless of the source, Vit D undergoes two steps of hydroxylation, as shown in Figure 1. The first hydroxylation step is carried out by both microsomal and mitochondrial enzymes in the liver. This can lead to the formation of 25OHD, which circulates in large amounts [21]. Most 25OHD is bound to serum proteins and generally has a half-life of less than three weeks. The second hydroxylation step occurs in the proximal tubule cells of the kidneys via 1-a-hydroxylase, resulting in the production of 25OHD2, also known as calcitriol. In addition to proximal tubule cells, this enzyme is also produced in many other cell types, such as cardiomyocytes, endothelial cells, and macrophages, but renal 1-a-hydroxylation is considered the major player in the circulation of calcitriol under normal conditions [22]. The metabolism of Vit D is discussed in detail in the Metabolism section of this review.
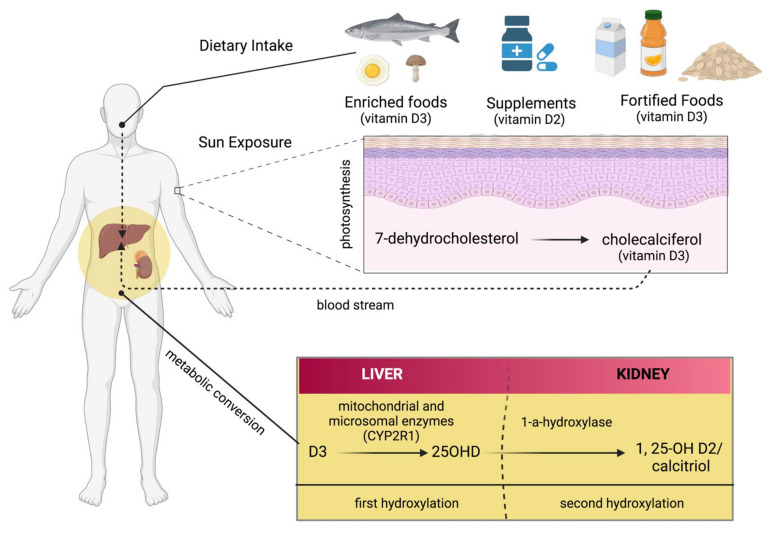



## 4. Metabolism of Vitamin D

The enzymes located in the endoplasmic reticulum (ER) (e.g., CYP2R1) or in the mitochondria (e.g., CYP24A1, CYP27B1, and CYP27A1) play an important role in the metabolism of Vit D. These enzymes are mixed-function cytochrome P450 oxidases (CYPs) that regulate Vit D metabolism via three main steps: 25-hydroxylation, 1α-hydroxylation, and 24-hydroxylation. In addition to the role of these three enzymatic steps in Vit D metabolism, we also discuss the enzymes 3-epimerase and CYP11A1, which play a minor role in this metabolic pathway [23,24,25].

Vit D is mainly produced in the liver. Early studies on 25-hydroxlase in the liver showed that the activity occurs in microsomal and mitochondrial fractions. Based on published data, some CYPs are currently known to have activity similar to that of 25-hydroxlase [26]. CYP27A1, for example, is widely distributed throughout the body, including the liver. It is considered the only mitochondrial hydroxylase that can perform a 25-hydroxlase-like function. However, this enzyme cannot form 25-hydroxylate of Vit D2. Another example of CYP with 25-hydroxlase-like activity is CYP2R1, which was found in the microsomal fractions of mouse livers [25]. In contrast to CYP27A1, this enzyme is able to 25-hydroxylate both Vit D2 and Vit D3. The circulating level of 25OHD in the blood can be considered as a useful marker for Vit D nutrition [27]. Other CYPs with similar activity to 25-hydroxylase are listed in Table 1. Among these enzymes with 25-hydroxylase activity, CYP2R1 appears to be the major player. However, other enzymes with 25-hydroxylase activity may also influence and/or contribute to both circulating and tissue levels of 25OHD [28].

In addition to the enzyme with 25-hydroxylase activity, other enzymes also play important roles in regulating vitamin D metabolism (see Table 1). The 3-epimerase enzymes are responsible for the inactivation of the major Vit D metabolites. For example, they convert 25OHD3 to 3-epi-25OHD3 in the liver. In addition, CYP27B1, which performs 1α-hydroxylase activity, converts 25OHD3 and 3-epi-25OHD3 to 1α,25(OH)_2_D3 in the kidneys. The epimeric forms of the primary Vit D metabolites, 3-Epi-25OHD3 and 3-Epi-1α,25(OH)_2_D3, exhibit low affinity with both DBP and VDR due to the changing orientation of the hydroxyle group. This conformational change leads to a decrease in calcium transport capacity, as well as decreased gene expression, in the human colon cancer cell line [29,30]. Vit D3 and Vit D2 can also be hydroxylated by the enzyme CYP11A1, leading to the formation of some new vitamin D metabolites: 20,22(OH)_2_D3 or 20,22(OH)_2_D2, and 20OHD3 or 20OHD2. 20,22(OH)_2_D3 and 22OHD3 are high in keratinocytes due to CYP11A1, suggesting that CYP11A1 can be activated by UVB exposure [23,31,32,33].

## 5. Overall Effects of Vitamin D: From Antioxidant to Genomic Effects

The mechanism of action of Vit D can be discussed in genomic and non-genomic terms. The genomic aspect has been studied in detail by Pike et al. [48] and Haussler et al. [49]. In the genomic mode of action, the vitamin D receptor (VDR) plays the key role. The vitamin D receptor is encoded by the *VDR* gene. VDR belongs to the superfamily of nuclear hormone receptors, which are ligand-induced transcription factors. It is worth noting that this receptor also functions as a receptor for the secondary bile acid, lithocholic acid. It consists of three main domains: first, a DNA-binding domain characterized by two zinc fingers that is able to bind specific DNA sequences (called VDRE); second, a C-terminal ligand-binding domain; and third, a specific region that is able to connet bowth domains [50]. Notably, VDR interacts with the retinoid X receptor. Interestingly, VDR binds to VDRE, forming a complex that differs from cell to cell by forming the action specificity of Vit D and thatcan regulate the expression of several genes, including C/EBPα and others [51,52]. In summary, the following aspects can be considered as the basis for the effect of *VDR/RXR* on target genomes: (a) the number of binding sites for the Vit D receptor depends specifically on the cell type; (b) the VDR/RXR heterodimer is considered the major active transcription unit, but not the only one; (c) the VDR binding sites are mainly (but not exclusively) classical hexamer half-sites separated by three base pairs; (d) the enhancers of the VDR-encoding gene are located near or far, i.e., promoterproximally or promoterdistally, and many enhancers are also located in clusters hundreds of kilobases away from their target genes; (e) enhancers are modular, and the binding sites they possess are binding sites for a number of different transcription factors; (f) these enhancers (populating a genome) are particularly dynamic and cell-type-specific. Vit D also has a rapid effect on selected cells that is probably unrelated to gene regulation and appears to be mediated by another receptor, which is potentially membrane-bound. One of the most important non-genomic effects of Vit D is the stimulation of calcium and phosphate uptake from the small intestine, known as transcalcification [53]. This term was coined by Norman’s laboratory to describe the rapid onset of calcium flow through the intestine of a Vit-D-fortified chick fed with 1,25(OH)_2_D [53]. Moreover, Vit D has been shown to modulate not only calcium, but also chloride channel activity, protein kinase C activation and distribution, and phospholipase C activity in various types of cell, including osteoblast, liver, muscle, and intestinal cells [54,55,56,57,58]. Vit D can also stimulate phosphate reabsorption in renal tubules and trigger the secretion of calcium from the bones into the blood. Another significant effect of Vit D administration is the reduction in pro-oxidative biomarkers and lipid peroxidation [57]; in fact, in patients with type 2 diabetes, Vit D administration reduces glucose-related pro-inflammatory proteins and 4-hydroxinonenal (a lipid-peroxidation marker involved in cardiovascular diseases, cancer, and neurodegenerative diseases) [58]. Notably, Vit D plays a key role in the homeostasis of neuron functions in humans. Recent meta-analyses correlate 25OHD serum levels with a high risk of cognitive impairment and memory decline [59,60]. Other studies correlate low 25OHD serum levels with the initiation and progression of Parkinson’s disease [61]. The conclusions of a meta-analysis on Vit D and neurodegenerative diseases declare that Vit D at 75 nmol/l could be useful to improve bone health, cognitive functions, and neuron survival through reductions in oxidative stress and improvements in mithochondrial metabolism [62].

## 6. Drug Interactions

The extent of the interaction of individual drugs interaction with Vit D is difficult to determine because Vit D activity can be influenced by the host-inherited genetic profiles of the previously mentioned VDR and CYPs genes [63,64]. In addition, the concomitant administration of CYP450 inducers such as rifampicin, isoniazid, barbiturates, and certain anticonvulsants (e.g., oxcarbazepine) may reduce the pharmacologic effects of Vit D analogs [65]. These agents are thought to induce the hepatic conversion of Vit D to inactive metabolites and have been shown to decrease circulating levels of active Vit D, sometimes accompanied by a decrease in serum calcium and an increase in parathyroid hormone levels. Patients receiving long-term anticonvulsant therapy have occasionally developed osteomalacia, presumably due to interactions between Vit D and calcium metabolism [66]. There have also been isolated reports of patients who responded poorly to Vit D supplements during treatment with phenytoin and/or primidone [67]. Some of the main interactions are listed in Table 2, below.

## 7. Genetic Factors Influencing Vitamin D Homeostasis

Vit D and its metabolites are transported to the liver, where they are hydroxylated and/or 3-epimerized, via DBP binding. CYP2R1 is the major actor in 25-hydroxylation metabolism, although several CYPs are involved in Vit D metabolism [23,25]. Thus, polymorphisms in the genes encoding these enzymes are thought to affect Vit D metabolism and homeostasis, although because several drugs and natural compounds have inducing or inhibitory effects on these enzymes, a true correlation between CYPs polymorphisms and serum Vit D levels is difficult to find. Several clinical studies have shown that people respond individually to Vit D3, with patients divided into high, moderate, and low responders [69,70,71,72]. Although several genome-wide association studies (GWASs) have identified about 700 Vit D target genes, it has not yet been explained which genomic and epigenomic variations affect Vit D homeostasis [64,73]. Mutations in *CYP2R1*, the major 25-hydroxylase of vitamin D, that affect CYP2R1 enzyme expression or function have been associated with rickets, low circulating 25(OH)D levels, and decreased sensitivity to vitamin D supplementation [74,75]. In addition, several SNPs in CYP2R1 (rs10832313, rs10766197, rs1562902, and rs10741657 in the 5′ promoter region, rs12794714 and rs1993116 in the 5’-UTR, and rs2060793 in the 3′-UTR) correlate with decreased circulating 25(OH)D levels and/or with common diseases, such as obesity, asthma, and multiple sclerosis, as well as cancer and all-cause mortality [76,77,78,79,80]. However, the question of how these variants affect CYP2R1 activity and lead to disease needs to be investigated further. Another genomic study focused on the VDR gene, which encodes the nuclear receptor VDR and is associated with 70% of the genetic effects on bone mass density and arthritis [81]. There are several polymorphic sites in the *VDR* gene recognized by the restriction endonuclease enzymes *Taq*I, *Bsm*I, *Apa*I, and *Fok*I: the alleles are called T-t, B-b, A-a, and F-f, respectively depending on the presence or absence of the restriction site. *Taq*I (rs731236) is located in exon 9 of the *VDR* gene, and consists in T > C substitution. The T nucleotide is also referred to as allele T, while the C nucleotide is defined allele t, and the polymorphism occurs in a CpG island, resulting in methylation. *Bsm*I (rs1544410) is located in intron 8 of the gene and consists in A > G nucleotide substitution, influencing transcript stability. The A nucleotide corresponds to allele B, and the G nucleotide corresponds to allele b. *Fok*I is a T > C nucleotide substitution on the *VDR* gene codon start (ATG → ACG) [82,83,84]. This shortened protein gains greater transcription activity due to the minor steric bulk of the molecule at the binding site of the transcription factor IIB [85]. The T nucleotide is referred to as allele f, while the C nucleotide is defined allele F. *Apa*I (rs7975232) occurs in intron 8 of the gene and the nucleotide C > A substitution is also referred to as A > a allele. The functional impact of this polymorphism has not yet been clearly described. These polymorphisms are strictly correlated to diseases and changes in homeostasis processes, such as bone mineralization and calcium intake [81,82,84]. For example, the VDR gene can interact with calcium in humans [83]. Currently, the functional significance of this polymorphism needs to be clarified. *VDR* genotyping via PCR-based methods is feasible and cheap (Table 3) [86].

There are two lesser-known polymorphisms in the 5′ promoter region of the VDR gene: *Cdx2* (rs11568820) and *GATA* (rs4516035). The *Cdx2* polymorphism is located upstream of exon 1 at the binding site of the transcription factor Cdx2 and consists of an A > G substitution, leading to the deletion of the binding site. The G allele is responsible for a 70% reduction in the transcriptional activity of the VDR. *GATA* polymorphism is located downstream of exon 1 at the GATA binding site, and, similar to rs11568820, the nucleotide substitution (T > C) results in the decreased promoter activity of the VDR. These SNPs are in linkage disequilibrium and are often analyzed as haplotypes [87,88]. These polymorphisms have been correlated with the risk of developing prostate cancer [89].

## 8. Vitamin D and Osteoporosis

Vit D3 promotes the absorption of dietary calcium via parathyroid hormone (PTH), contributing to adequate calcium homeostasis [90]. The activity of 25(OH)D-1α-hydroxylase, the enzyme responsible for the conversion of 25(OH)D to 1,25(OH)_2_D, is stimulated by PTH and inhibited by 1,25(OH)_2_D itself [91]. Furthermore, Vit D3 suppresses the activity of PTH and its secretion by inhibiting the proliferation of parathyroid cells. This active calcium absorption occurs through the induction of the synthesis of a protein expressed at the brush borders of intestinal epithelial cells, which bind the ion and transport it from the lumen to the cell cytoplasm. In addition, 1,25(OH)_2_D also facilitates the passive absorption of calcium by increasing the permeability of intercellular “tight junctions” [92]. Vit D deficiency results in a compensatory increase in PTH, which in turn stimulates bone turnover and renal tubular reabsorption of calcium to maintain calcium homeostasis [93]. Calcium and phosphorus are deposited in the collagen matrix and form hydroxyapatite, which gives strength to the skeleton. Through these mechanisms, Vit D acts indirectly on the bones (see Figure 4) [94].

Vit D deficiency leads to hypocalcemia and hypophosphatemia, which classically cause rickets in children and osteomalacia in adults. Both diseases are caused by impaired bone mineralization due to an inadequate calcium-phosphate product and the effect of PTH on the kidneys, causing phosphaturia [95]. Adequate levels of Vit D have an important effect on bone mass in young and old people [96]. Since bone mineral density directly correlates with fracture risk, Vit D is crucial for the treatment of osteoporosis [97]. During life, the skeleton is subject to different processes, which can be termed modeling and remodeling. Modeling aims to adapt the structures of the bones to the mechanical stresses caused by body growth and age. Remodeling continues throughout life, with the aim of replacing damaged or aged bone tissue with new tissue, without altering the affected surface, ultimately controlling bone structure and function [98]. The cells that regulate these processes are osteoblasts, osteoclasts, osteocytes, and lining cells. Osteoblasts originate in multi-potential mesenchymal progenitors and are exclusively responsible for the formation, deposition, and mineralization of bone tissue [99]. They also control the recruitment, differentiation, and maturation of osteoclasts, which participate in resorption activity. In addition, osteoclasts associated with bone resorption also express several factors that regulate osteoblast function [100]. Osteocytes, which are terminally differentiated osteoblasts, act as mechanosensors and modulate both osteoblast and osteoclast activity. Similarly, bone lining cells play a role in assisting osteoclasts and osteoblasts in bone remodeling [101]. The discovery of osteoprotegerin (OPG), a receptor activator of nuclear factor kappa-B ligand (RANKL) derived from osteoblasts, led to a better understanding of the mechanism of cross-communication between osteoblasts and osteoclasts. RANKL is a transmembrane protein on the surface of osteoblast cells that binds to its own receptor, RANK, which is located on the surfaces of both osteoclast precursors and mature osteoclasts [102]. This cell-to-cell contact, in combination with m-CSF, which is also produced by osteoblasts, stimulates the differentiation of precursor cells into osteoclasts, and increases their activity [102]. 1,25(OH)_2_D regulates this process via the VDR receptor expressed by the osteoblast by inducing RANKL [103].

Osteoporosis is a clinical condition characterized by a progressive reduction in bone mass and a concomitant alteration in the skeletal microarchitecture, leading to a loss of bone strength, which increases the risk of fractures due to very mild trauma. Osteoporotic fractures mainly occur at the level of the vertebral bodies and the femoral neck [104]. There are essentially two forms of the disease: primary and secondary osteoporosis. Primary osteoporosis is in turn divided into Type 1 and Type 2: (a) postmenopausal osteoporosis (Type 1) occurs in women within 15–20 years of menopause and is thought to result from factors related to or exacerbated by estrogen deficiency [105]; (b) age-related osteoporosis (Type 2) occurs in men and women over 75 years of age and may be more directly related to the aging process. Secondary osteoporosis encompasses a broad range of conditions related to diseases and medication that affect skeletal metabolism [105]. In this context, adequate Vit D levels and dietary calcium intake contribute to achieving maximum bone mass, maintaining bone mineral density (BMD), and compensating for the loss of calcium associated with the increase in bone turnover during menopause or with the decrease in calcium intake in old age [106]. The muscle weakness found in hypovitaminosis D contributes to the risk of falling, which in turn increases the risk of fracture [107]. Serum 25OHD is a robust and reliable marker of Vit D status, and although there is no consensus on the definition of optimal serum 25OHD levels, Vit D deficiency is defined by most experts as a serum 25OHD level <50 nmol/L (<20 ng/mL). The Institute of Medicine (IOM) considers 25OHD levels > 50 nmol/L (>20 ng/mL) as sufficient, while other scientific societies, such as the Endocrine Society, advocate broader ranges. With this in mind, a serum 25OHD level of >75 nmol/L (>30 ng/mL) is considered normal, and a level of 50–75 nmol/L (20–30 ng/mL) is considered Vit D insufficiency [108]. Adequate levels of Vit D have an important effect on bone mass in young and old people. Hypovitaminosis D negatively affects calcium metabolism, osteoblastic activity, matrix ossification, bone remodeling, and, thus, bone density [109]. Vitamin D plays a fundamental role in the regulation of skeletal metabolism, and its target organs are the intestines and bones themselves [110].

## 9. Vitamin D and Muscle Homeostasis

Low levels of Vit D are associated with an increased risk of falls and proximal weakness. Muscle pain and hypotonia are typical clinical features of rickets and osteomalacia. Proximal myopathy with unsteady gait is observed in adults with severe chronic Vit D deficiency [111]. Observational studies have reported an association between low Vit D levels and the risk of falls in the elderly. 1,25(OH)_2_D stimulates muscle protein production and contributes to muscle contraction by regulating the mechanism of calcium transport at the level of the sarcoplasmic reticulum [112]. Age-related sarcopenia is associated with increased risk of falls, disability, and mortality in the elderly [113]. VDR’s expression in skeletal muscle suggests that Vit D may act directly on this tissue. Experimental studies in skeletal muscle-specific VDR knockout mice suggested a Vit D-specific effect on muscle fiber size, as a reduction in muscle fiber diameter of the II type was observed in these animals [114]. This provides biological evidence for a direct role of Vit D deficiency in sarcopenia. This is also supported by another study speculating the possibility that Vit D modulates myostatin, a negative regulator of muscle mass [115]. During aging, Vit D synthesis in the skin mediated by UV radiation decreases, as well as the renal activation of 25OHD and VDR concentrations in muscle tissue. The combination of all of these effects could make muscle more susceptible to Vit D deficiency and promote falls.

Adequate sunlight exposure is the most cost-effective method to maintain Vit D levels because its natural sources in food are very limited [116]. It has been calculated that whole-body sun exposure in summer provides the equivalent of 10,000 IU Vit D. Studies have demonstrated that, on average, 25OHD levels increase by 0.5 to 1 ng/mL for every 100 IU of Vit D supplement administered [117]. Much higher doses are likely to be required in obese individuals or those with malabsorption (including after bariatric surgery). People with darker skin, older age, and higher BMI are more likely to be affected by Vit D deficiency [118,119].

## 10. Vitamin D Supplementation and Modulation of Immune Functions: Putative Implications for Cancer Patients

Vit D is able to modulate immune functions in humans [120]. Both 25(OH)D and 1,25(OH)_2_D acts in multiple ways in several immune cells, such as macrophages, monocytes, B and T-type lymphocytes. Vit D is able to modulate the expression of the genes involved in innate and adaptative immune functions. In fact, epidemiological studies correlate autoimmune diseases and the risk of infection with low serum levels of 25OHD [121]. However, interventional studies aimed at improving the serum levels of 25OHD in patients with immune-related diseases are heterogeneous and not always significant [122]. Historical evidence for the links between Vit D and innate immunity comes from reports dating from the mid-18th to the early 19th century, before the antibiotics era, when Vit-D3-rich cod liver oil and sunlight exposure were used to treat tuberculosis [122]. Humans have a precise system of regulation for their immune activities based on the endogenous hyperactivation of Vit D. During an infection process, activated polymorphonuclear granulocytes, through pro-inflammatory chemokines and growth factors, stimulate CYP27B1, which is able to convert 25OHD to 1,25(OH)_2_D [123], which, through autocrine mechanisms, increases the expression of cathelicidin. Cathelicidin exerts antiviral and antibacterial effects [124] on Mycobacterium tuberculosis and others (Figure 3). Moreover, 1,25(OH)_2_D acts in a paracrine manner. This stimulates adjacent macrophages, which often appear in the bloodstream, reaching serum concentrations of 30 ng/mL, causing hypercalcemia (hypercalcemia is also a marker of infectious status) [125]. Moreover, 1,25(OH)_2_D is able to maintain immune tolerance in APC cells and finely manage the surface expression of MHC class II, immunogenic cytokines, and co-stimulation molecules (Figure 5) [126]. The modulation of the immunogenic cytokine profile is a key orchestrator of immune homeostasis; for example, Vit D increases IL-10 expression, which is characterized by anti-inflammatory activity [126]. Conversely, 1,25(OH)_2_D reduces the expression of the pro-inflammatory and atherogenic cytokines involved in immune hyperactivation, such as IL-6 and IL-17 [127]. Notably, 1,25 (OH)_2_D is able to modulate the metabolism and immunogenic activity of NK cells; although the clinical studies are very heterogeneous, it would seem that 1,25 (OH)_2_D is able to activate NK cells and, consequently, offers the potential for use in cancer patients subjected to treatment with immune check-point inhibitors [128].

## 11. Vitamin D and Cancer

### 11.1. Association between Vitamin D and Risk and Prevention of Cancer

The administration of Vit D3 during oncological treatment quickly became prominent in day-to-day medical practice. Vit D3 is widely used in neoadjuvant, adjuvant, and first-and-beyond lines of treatment, in relation to the comorbidities that patients present [129]. Pre-clinical studies (based on laboratory and animal analyses) suggest mechanisms through which Vit D3 may inhibit carcinogenesis and slow tumor progression through different pathways, and observational studies suggest that Vit D3 could confer greater protection against cancer mortality than cancer incidence, notwithstanding reductions in both [130,131,132]. Oncology patients have an intrinsic tendency toward comorbidity: patients with cancer are prone to have one or more diseases (i.e., cardiopathy, diabetes, hypertension, osteoporosis) that are, at least apparently, unrelated to the concurrent neoplasm [133]. One of these clinical conditions, typically identified during random blood tests, is Vit D3 deficiency; its serum level reduction is widely correlated to osteoporosis, fractures, falls, and more general bone remodeling alterations. According to Rusinska et al., patients should be categorized according to age, comorbidity, and serum level of Vit D3 deficiency [8]; adults whose serum levels are below 50 nmol/L should be treated through behavioral education (sunbathing 10.00–15.00 during May–September; due to the decreased efficacy of skin synthesis, this is not recommended in older people (65–75 yo) or people with dark complexions) and oral supplementation (800–2000 UI/day, according to weight and dietary Vit D3 intake) [134]. Oncologists and General Practitioners (GP) should interpret these blood test results and act accordingly to prevent Vit-D3-related symptomatic manifestations [135].

### 11.2. Pathophysiology of Vitamin D in Cancer Prevention

Despite its role in bone pathology, Vit D3 has also been investigated in other clinical conditions, namely cardiovascular disease and cancer development to determine whether it plays a protective role, t, with observational studies correlating low serum 25(OH)D levels with an increased incidence of the aforementioned pathologies. However, these studies failed to demonstrate causality, so this question remains unanswered [136]. JoAnn E. Manson et al. designed a randomized, double-blind, placebo-controlled, 2 × 2 factorial trial to examine the benefits and risks of Vit D3 and marine omega-3 fatty acids for the primary prevention of cancer and cardiovascular disease among 28,871 men aged ≥50 and women ≥55 [137]. The patients were randomized and enrolled in four different groups: Vit D3 supplement, Omega 3 fatty acid supplement, both active agents, or both placebo; the groups were homogeneous in composition, according to age, sex, and race. None of the patients had cancer (except for non-melanoma skin cancer) or cardiovascular disease history when they entered the study [137]. A total of 1617 participants developed the primary endpoint of total invasive cancer, with similar events in the Vit D3 and placebo group, and the incidence of site-specific cancer (prostate, breast, colorectum) did not differ significantly between the groups; during the follow-up, 154 participants in the Vit D3 subgroup died from cancer versus 187 in the placebo group [138]. In conclusion, while the study had sufficient size and duration (≥5 years) it could n-t demonstrate a valid impact of daily Vit D3 supplementation on cancer incidence reduction in healthy American adults, even though it suggested a new factor to be taken in consideration, BMI, since the normal-weight participants experienced a possible treatment-associated reduction in incidence).

### 11.3. Vitamin D levels and Breast Cancer

It has been shown that epithelial breast cells possess the same enzyme metabolism system as the kidneys, rendering the effect of Vit D3 on breast cancer plausible: the control of the cell cycle is supported by the chemopreventive actions of 1,25(OH)_2_D or calcitriol, the active form of Vit D3. Furthermore, preclinical and ecological studies showed that the implementation of Vit D3 induces differentiation and apoptosis while inhibiting cell proliferation [139,140]. A preclinical study was conducted to investigate this hypothesis. The researchers incubated human mammary cells with physiological concentrations of 25(OH)D and discovered that these cells produced sufficient amounts of 1,25OHD to inhibit cell growth [141]; however, the concentration that would have been required to mediate this effect in vivo would have been well above the physiological range. Thus, the authors suggested that the presence of Vit D and its core receptorial and metabolic complexes in breast cells might act in an autocrine or paracrine manner [141]. A meta-analysis evaluated 9 studies, aiming to assess the interactions between level of Vit D and postmenopausal breast cancer; 5206 patients and 6450 controls were included. demonstrated A non-linear inverse association between circulating 25(OH)D and postmenopausal breast cancer (BC) development risk was demonstrated [142]. These findings may be supported by the fact that menopausal women experience several hormonal and physical changes, including weight gain and obesity, which promote an increase in circulating estrogen and imply an increased risk of hormone-dependent breast cancer [143]. Supplementation with Vit D can reduce the risk of this pathological development, since it can downregulate the expression of estrogen receptors and reduce the synthesis and signaling pathways of these hormones [144].

### 11.4. Vitamin D and Prostate Cancer

Several studies tried to assess the relation between Vit D, its bioactive form, calcitriol, or its analogues, and prostate cancer. Calcitriol, 1alpha-hydroxiVit D2 (doxercalciferol), and 19-nor-1alpha-25-dihydroxyVit D2 (paricalcitriol) were evaluated as single agents in patients with castration-resistant and castration-sensitive disease [145]. Although a decrease in the rate of PSA in castration-sensitive disease and some evidence of activity in castration-resistant prostate cancer (CRPC) (19% PSA response rate) were detected, none of these studies provided convincing evidence of clinically important single-agent activity [146]. The analysis could have been influenced by the concomitant use of Vit D analogues and corticosteroids, nominally dexamethasone, due to its well-known ability to control the hypercalcemic effect of calcitriol and its demonstratable antineoplastic activity [147]. When comparing analogues to single-agent glucocorticoids in large clinical trials, PSA response rates have been reported in the range of 3–10%. Furthermore, single-agent doxercalciferol was studied in CRPC (12.5 mg per os) among 26 patients: it did not reach statistical significance, with no substantial decrease in PSA; paricalcitol was evaluated on a 3-times-per-week schedule and, while it was very well tolerated and safe to use, it did not demonstrate a serum PSA reduction [147,148]. A combination of Vit D3 and cytotoxic agents was also explored through two different randomized trials. The first (ASCENT I) compared the PSA response rates in 250 patients randomized to receive standard therapy for CRPC (docetaxel 36 mg/m^2^ weekly, every 4 weeks, and every 6 weeks) compared to the same therapeutic schedule with the addition of calcitriol (DN-101), at 45 mcg per os daily [149]. While the primary endpoint was not achieved, there was a substantial numerical difference in favor of the DN-101 arm; the patients treated with calcitriol also experienced a greater level of median survival (24.5 months vs 16.4 months). A larger trial was then initiated (ASCENT II) to determine the apparent survival advantage of DN-101 plus docetaxel. Unfortunately, the results did not demonstrate that calcitriol enhanced the antitumor efficacy of docetaxel in CRPC. The study was stopped after an interim analysis showed that the survival was statistically worse in the DN-101 arm [149,150].

### 11.5. Vitamin D and Melanoma

One of the neoplasms that can be most intuitively related to vitamin D status or alterations in vitamin D metabolism is skin cancer. As early as 30 years ago, 1,25(OH)_2_D3 showed anti-melanoma activity, with effects on differentiation, apoptosis, malignant cell invasion, and metastasis, at least in vitro [151,152]. Researchers have shown that the same mechanisms involved in breast carcinogenesis discussed above (the expression of VDR in cancer cells and its possible influence on the cell cycle) also play a role in this oncological entity [151]. Another study was conducted to assess the in vivo activity. It showed that Vit D3 suppressed the growth of human-malignant-melanoma-derived xenografts in immunosuppressed mice [152]. There have been several studies attempting to investigate the effects of serum Vit D3 levels on cutaneous malignant melanoma (CMM) at the time of diagnosis and/or during the follow-up period after treatment. However, the results were not consistent with each other. Newton-Bishop et al. indicated that high Vit D3 levels at the time of diagnosis might have a protective effect, while other authors demonstrated that this protection was explained by the variability in the 25(OH)D3 levels during follow-up and not by the levels at the time of diagnosis per se [153,154]. Another group attempted to summarize and improve these important biological concepts in a double-blind placebo controlled phase III study, which unfortunately has not yet been published (NCT01748448) [155]. De Smedt et al. aimed to enrol 500 patients aged 18–80 years with histologically proven malignant melanoma at high risk of recurrence, namely stage IB–III, and to determine whether Vit D supplementation after the tumor removal had a protective effect on CMM recurrence [156].

### 11.6. Vitamin D and Colorectal Cancer

The association between Vit D3 and the development of colorectal cancer (CRC) has been widely covered in the literature, proposing an inverse relation between 25OHD serum levels and CRC risk [157]. The main sources of information are four different meta-analyses, which, unfortunately, almost entirely refer to the same nine prospective studies. Each manuscript, however, focuses on a different peculiarity of this relation, with each article demonstrating the inverse relation between the risk of developing CRC and 25OHD serum levels [157,158,159,160,161]. Lee et al. [158] showed a stronger influence of said level on rectal cancer while Gandini et al. and Touvier et al. [159,161] both highlighted a reduction in cancer incidence per 100 UI/L and 25 nmol/L increase in circulating Vit D3, respectively. A later meta-analysis by Maalmi et al. analyzed five prospective cohort studies from three different geographic areas (USA, Europe, and Japan) and demonstrated consistent results with the above studies, revealing a significant reduction in all-cause and CRC-specific mortality (up to 35%) [160]. In patients with higher Vit D3 levels, the results were highly statistically significant, although the sample size of the study was limited: only one of them reached 1202 included patients from a total population of 2330. Maalmi et al. elaborated further on their research in a recent update. They were able to include a total of 11 prospective cohorts and reached the same results as before. A subgroup analysis showed that the European studies, which happened to have larger sample sizes and included patients in stage I–IV, showed a clearer association between 25OHD serum levels and overall survival (OS) [160]. A Phase II, double-blind, randomized clinical trial (RCT), NCT0151621645, enrolled patients from April 2012 to November 2016, comparing previously untreated metastatic CRC randomized to FOLFOX-bevacizumab plus high-dose Vit D3 (8000 IU/day for 2 weeks followed by 4000 IU/day) versus FOLFOX-bevacizumab plus standard-dose Vit D3 (400 UI/day) [162]. The analysis performed, despite not reaching statistical significance, demonstrated a better PFS in patients treated in the high-dose Vit D3 arm.

### 11.7. Vitamin D and Head-and-Neck Cancer

Head-and-neck cancer (HNC) primarily affects the elderly, in whom metabolism and homeostasis appear to be impared at various levels, either because of comorbidities or intrinsic frailty. A recent publication has shown that the prevalence of Vit D3 deficiency is higher in HNC patients [163]. Pu et al. performed a meta-analysis evaluating Vit D3 exposure and HNC incidence and mortality, and the possible effects of *VDR* polymorphisms [164]. They reported an inverse association between the two aforementioned parameters and dietary or supplemental Vit D3 intake, which improved prognosis and quality of life [165].

### 11.8. Vitamin D and Bladder Cancer

Bladder cancer (BLC) and its relationship with Vit D3 serum concentration have been investigated in recent years. Baykan et al. performed a genetic analysis aiming to demonstrate a correlation between *VDR* polymorphisms and the risk of developing BLC in a Turkish population; they compared Vit D3 levels and VDR polymorphism in 101 cancer patients and 109 control subjects [166]. The investigators demonstrated a statistically significant difference in the genotype distribution of *Fok*I polymorphism, which was unfortunately lost when the odds ratio (OR) was adjusted by smoking status. *Fok*I polymorphisms have also been investigated in other smoking-related cancers, such as lung cancer, in a recent meta-analysis by Duan et al. [167], emerging as a risk factor; there is, however, no firm demonstration of a causal connection between smoking and gene alteration. A statistically significant inverse association between low 25OHD serum levels and increased risk of bladder carcinoma development was highlighted in two different meta-analyses [168,169]. Zhang et al. reported that patients with Vit D deficiency had a higher risk of developing bladder cancer compared with patients with higher 25OHD serum levels [168]. On the other hand, Zhao et al. stated that high 25OHD serum concentrations play a crucial role, decreasing bladder cancer risk by 60% [169].

### 11.9. Vitamin D and Onco-Hematological Cancer

Exposure to sunlight has been studied as a possible protective factor in various hematologic malignancies [170]; therefore, Vit D3 may play a role in their prevention. A recent meta-analysis included 30 case-control or cohort studies aiming to assess whether Vit D3 intake via diet and supplements or their surrogates (sunlight/UVR exposure) could be a valid protective factor in non-Hodgkin lymphoma (NHL) [171]. The analysis showed a protective effect of UVR exposure on NHL; furthermore, serum Vit D3 levels below 25 nmol/L were correlated with a higher risk of developing NHL and related subtypes, although these results were not significant. Currently a Phase III, double-blind, interventional RCT (“Indolent Lymphoma and Vit D—ILyAD Study”, NCT03078855) is investigating whether there may be a potential progression-free survival (PFS) benefit in patients with NHL treated with oral Vit D3 2000 UI/day plus rituximab compared to placebo plus rituximab [172]. The study, at the time of writing, is still in the recruiting phase. Drake et al. designed a retrospective, observational study in which they demonstrated a poorer prognosis in newly-diagnosed NHL patients who had low serum Vit D3 levels; moreover, supplementation appeared to be required in some histotypes (particularly T-cell lymphomas), even in the optimal range [173]. However, their manuscript failed to establish a valid causal relationship between low serum Vit D3 levels and a less favorable prognosis.

## 12. Vitamin D and Cardiovascular Diseases

Vit D receptors are widely expressed in the human body, including in cancer cells and the liver, kidneys, and cardiovascular system [174]. Recently, it was confirmed that VDRs are expressed both in vascular endothelial cells and in cardiomyocytes, modulating their mitochondrial metabolism, calcium homeostasis, and the production of endocardial metalloproteases [175]. A recent publication found that low serum levels of 25OHD are associated with a high risk of cardiovascular diseases, including stroke, heart failure, and general cardiovascular mortality [176]. A recent meta-analysis confirmed, however, that Vit D supplementation does not reduce adverse cardiac events, either in elderly or in non-elderly patients. In addition, it was demonstrated that exogenous Vit D supplementation does not reduce cardiovascular mortality, heart attack, MACE, or myocardial infarction [177]. However, it is assumed that Vit D supplementation in pediatric patients may reduce cardiovascular mortality in adulthood, possibly due to epigenetic modifications that improve cardiovascular health. Furthermore, observational studies confirmed, within their secondary objectives, the existence of an inverse association between serum Vit D levels and the incidence of heart diseases, although several confounding factors may alter the interpretation of the data and a conclusion cannot be unequivocally determined [178,179]. The primary outcomes were Vit D supplementation and the risk of osteoporosis in postmenopausal women and the reduction in the risk of dialysis in subjects with renal insufficiency, respectively. It is therefore important to emphasize that these studies always involved older people, or, in any case, people not of premenopausal age; therefore, the effective long-term benefit of Vit D for cardiac events was not included as a primary outcome. Consequently, further studies of cardiovascular outcomes should be designed, using both a younger population and higher doses of Vit D. Another recent study confirmed that the early administration of high-dose enteral Vit D provided no benefit over placebo in terms of mortality or other nonfatal outcomes in critically ill patients with Vit D deficiency [180]. By contrast, other studies indicate that daily Vit D supplementation at 2000 IU for three months improves endothelial function and reduces the expression of oxLDL and ICAM1 (cardiovascular risk factors) in patients with hypertension and type 2 diabetes [181]. Another, similar, study showed that the daily administration of 2000 IU and 800 IU of Vit D lowered mean systolic blood pressure over an observation period of two years [182].

## 13. Vitamin D and Human Microbiota

The human microbiota is the community of commensal, symbiotic, and pathogenic microorganisms that a human harbors. It consists of about 10–100 trillion microbial cells of bacteria, archaea, fungi, algae, and small protists [183]. The collection of genes harboring these microorganisms forms the human microbiome [184]. The most diverse, abundant, and well-studied human microbiota is that of the gut, but the microbiota of the skin, lung, oral cavity, and genitourinary tract are also proving to be important features in human pathophysiology [185]. Due to the emerging importance of the microbiota for human health and behavior, the human microbiota is now even considered an additional “organ” [186]. The gut microbiota comprises four main phyla: *Firmicutes, Bacteroidetes, Actinobacteria*, and *Proteobacteria*, the first two of which are found in abundance in human feces. Genetic and non-genetic factors influence the composition of the microbiota, resulting in a dynamic environment. However, diet is responsible for most of the changes in the composition of the gut microbiota [187,188]. Several pieces of evidence indicate that the human microbiota affects the production and degradation of nutrients and human metabolites, such as Vit D and its active metabolites. Indeed, Vit D has been shown to both affect and be affected by the gut microbiota: the latter altered intestinal Vit D metabolism, while probiotic supplements affected circulating 1,25(OH)_2_Dlevels [173,189,190,191,192,193]. Moreover, Vit D has been linked to a wide range of biological activities, including the modulation of gut mucosal immunity and the integrity of the intestinal barrier and mucus layer, and it is mediated by VDR. VDR is highly expressed in normal gut epithelial cells, especially in the crypts. The VitD/VDR signaling pathway plays a major role in maintaining gut homeostasis through the regulation of tight junctions and adherent junction components, as well as the release of antimicrobial peptides, such as defensins [194,195,196].

In this context, a dynamic interplay between active Vit D metabolites and butyrate-producing phyla, such as *Firmicutes*, has been reported [197,198,199]: this result was possibly achieved because butyrate-producing gut microbiota stimulate the local production of active Vit D (i.e., 1,25(OH)_2_D) through colon-resident immune cells, including dendritic cells. *Firmicutes*, along with other butyrate-producing bacteria (*Coprococcus* and *Faecalibacterum*), have been shown to influence VRD signaling, thereby promoting anti-inflammatory activity [200]. In addition, the modulation of such bacteria as *Firmicutes* involves mucus-controlling genes, and butyrate increases the healing processes occurring in the gut [191]. An observational study on 567 older men highlighted that high levels of 1,25OHD were associated with butyrate-producing bacteria and, consequently, improved gut microbial status [197]. Two interventional studies on men and women with Crohn’s disease and ulcerative colitis showed significant changes in the composition of the gut microbiota after the administration of Vit D3 [201,202]. Several human studies that manipulated the gut microbiota by introducing *Lactobacilli* reported increased Vit D receptor expression in human cell cultures [203]. Similarly, a GWAS analysis of the gut microbiota from 1812 individuals controlled for diet and non-genetic parameters identified variations in VDR [204]. The gut-microbiota–Vit-D axis also works the other way around: it has been reported that intact Vit D signaling is important for a healthy gut microbiome: mice with disrupted Vit D metabolism had intestinal dysbiosis [190], while adults with cystic fibrosis supplemented with Vit D modified their gut and airway microbiota, with an enrichment of *Lactococcus*, which is associated with improved gut health [205]. Vit D also restored high-fat-diet-induced gut microbiota dysbiosis by increasing the relative abundance of *Lactobacillii* and attenuated high-fat-diet-induced hepatic injury in vivo [195]. By contrast, high doses of Vit D given to healthy adults did not change the composition of the stool microbiome [206]. These human studies suggest that Vit D supplementation provides benefits to the gut microbiome only when its levels are physiologically low, and not in healthy individuals with normal levels of Vit D.

## 14. Vitamin D and Overall Risk of Infection

Vit D also deeply influences the immune system, by acting directly on immune cells to promote an anti-inflammatory state; Vit D deficiency encourages the development of a proinflammatory state [207]. Accordingly, there is solid evidence that Vit D supplementation can reduce the rate of infection. For example, it has been shown that Vit D status can impact the development and severity of acute respiratory tract infections, such as influenza and pneumonia [208,209].

A recent meta-analysis determined that Vit D3 supplementation significantly decreases the risk of developing acute respiratory tract infections: supplementation was especially effective in participants who presented low serum levels of 25OHD at baseline [210]. Vitamin D has also been associated with disease severity in COVID-19 patients. Those who presented Vit D deficiency on admission had a greater risk of both requiring invasive mechanical ventilation and mortality [211,212]. Patients with COVID-19 were found to have a higher prevalence of Vit D deficiency than control groups [212]. Vit D has recently been demonstrated to increase the expression of ACE2, the main receptor of SARS-CoV-2, which facilitates SARS-CoV2 host-cell invasion in vitro. Higher concentrations of this receptor can also promote a greater proinflammatory response, which may exacerbate organ damage in patients [212]. Indeed, one small Italian study established a possible increased risk of in-hospital mortality due to Vit D supplementation [213].

## 15. Conclusions

We conducted a systematic review of both observational and interventional studies to comprehensively evaluate the effects of vit D intake on chronic diseases, particularly cardiovascular disease and cancer. Vit D is a fat-soluble vitamin that acts as a steroid hormone, and its primary source is the UVB-induced conversion of 7-dehydrocholesterol to Vit D in the skin. Secondary sources include Vit D-containing foods and dietary supplements [12,15]. Regardless of the source, Vit D undergoes multiple hydroxylation reactions to its active form, 1,25-dihydroxyvitamin D, which can exert biological effects. Vit D activity is recognized as part of the endocrine system, maintaining extracellular calcium levels via the regulation of both intestinal calcium absorption and osteocyte turnover. Most of the biological activities of this steroid hormone are exerted by binding to the vitamin D receptor. VDR is one of the nuclear receptors for steroid hormones that functions as a ligand-activated transcription factor, thereby regulating gene expression. Although VDR is expressed primarily in the small intestine, bones, and kidneys, organs that are sensitive to vitamin D because of its central role in calcium homeostasis, this receptor has also been found in other tissues and organs, including the skin, and some cell types in the immune system, suggesting a possible influence of Vit D on the immune response to various diseases [114]. It is well known that childhood rickets is a direct consequence of severe vitamin D deficiency in the diet. However, the association between vitamin D deficiency and osteomalacia in adults is more controversial. Daily supplementation with 400 IU of vitamin D may prevent rickets and osteomalacia when serum 25OHD levels rise above 30 nmol/L, which is considered the threshold for vitamin D deficiency [214]. In any case, supplementation with Vit D alone seems not to be the solution for reducing fracture risk. A combination of calcium and vitamin D supplementation (1000 mg and 800 UI daily respectively) can reduce the incidence of hip and other fractures in adults by 20% by both increasing 25OHD serum levels by 25 nmol/L and improving BMD and bone quality [215,216,217]. Muscle homeostasis is also related to Vit D, which affects muscle development and strength. Muscle weakness is typical in individuals with chronic renal dysfunction due to the depletion of 1,25(OH)_2_D and in individuals with genetic alterations in *CYP27B1* and *VDR* [64,73]. Vit D deficiency is also associated with an increased risk of infections and immune and autoimmune diseases. In particular, there is a strong association between low serum levels of 25OHD and the risk of developing multiple sclerosis. Considering the role of 1,25(OH)_2_D in the downregulation of the adaptive immune system, it is conceivable that Vit D deficiency promotes autoimmune diseases, such as multiple sclerosis, inflammatory bowel disease, and type 1 diabetes [6]. Current knowledge of vitamin D status and cardiovascular disease risk indicates that there is no clear benefit of vitamin D supplementation in individuals at risk of heart failure, myocardial infarction, and stroke. However, few data are available to draw a conclusion, and specific studies are needed to better elucidate the role of vitamin D supplementation in individuals with cardiovascular disease. Data support the association between low serum Vit D levels and type 2 diabetes mellitus and metabolic syndrome, especially the association between the slowing of the progression from prediabetes to T2DM and Vit D supplementation [218]. Recently, the role of the microbiota in human health has become increasingly important, and several studies are underway that aim to better understand the relationship between the composition of the gut microbiota and the development of disease in children and the elderly. Studies both in vivo and in humans support the benefits of Vit D supplementation for gut microbiota composition, although some questions remain about the proper 25OHD serum level to improve the microbiota–Vit-D axis [206,219]. In the context of the COVID-19 pandemic, the relationship between vitamin D status and severity of SARS-CoV disease has received attention. Observational studies and meta-analyses associated low serum Vit D levels with both high COVID-19 mortality and morbidity, with many confounding factors, including a similarity between the risk factors for Vit D deficiency and for COVID-19 [210,211]. However, recent studies suggest that Vit D supplementation is of no benefit in hospitalized patients with COVID-19 and that there is no association between genetic serum 25OHD levels and the risk of hospitalization in COVID-19 patients [220,221]. Thus, it is difficult to draw a definitive conclusion about the role of Vit D in reducing the risk of severe SARS-CoV-2 infection. There are strong preclinical and epidemiological studies linking Vit D to cancer risk, proliferation, and prognosis. The role of Vit D in carcinogenesis appears to be multifactorial, suggesting a possible role in the control of cellular processes, such as inflammation, proliferation, angiogenesis, differentiation, invasion, and apoptosis. Altogether, epidemiologic studies are still inconclusive in determining the true effect of vitamin D on reducing cancer risk and improving patient outcomes [137,222,223]. New, adequately designed RCTs must be performed. However, it can be said that supplementation with Vit D and its analogues is a promising strategy for both the prevention and treatment of cancer. Moreover, the use/support of Vit D in a particular setting of patients, such as those affected by HIV infection and cancer, should be mandatory, with the aim of reducing the risk of opportunistic infections, mainly during anticancer treatment. For patients, an adequate serum value of Vit D should reduce the risk of cancer and immunity deficiency [224,225,226,227,228]. Most importantly, a recent meta-analysis revealed that the oral administration of Vit D reduces plasma levels of MDA, a marker of lipid peroxidation [218,219], at doses of 100,000 and 200,000 IU per month. The antioxidant mechanisms of Vit D are related to its membrane and lipoprotein scavenger activities [220,221,222], which can reduce iron damage, ferroptosis, and ROS production [223,224,225].

## 16. Perspectives

This paper aims to draw the attention of the scientific community to the use of Vit D in clinical practice in patients with and without chronic diseases. To this purpose, we collected available data on the multiple effects of Vit D administration in particularly vulnerable patients, such as those with cancer, as well as those with and without cardiovascular diseases. The observations described here could significantly contribute to the knowledge of Vit D and its beneficial effects. Rigorous placebo-controlled trials based on the daily administration of Vit D in patients with cancer and/or cardiovascular diseases or patients treated with drugs affecting bone or muscle homeostasis should be evaluated.

## Figures and Tables

**Figure 1 antioxidants-11-01090-f001:**
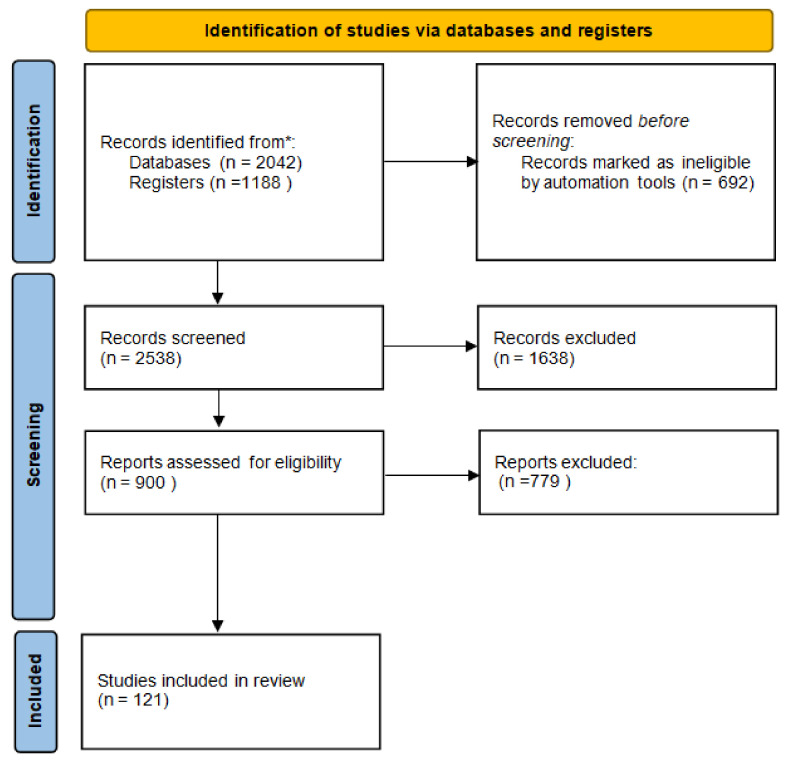
Flow diagram of systematic review. * The databases were last accessed on 31 January 2022.

**Figure 3 antioxidants-11-01090-f003:**
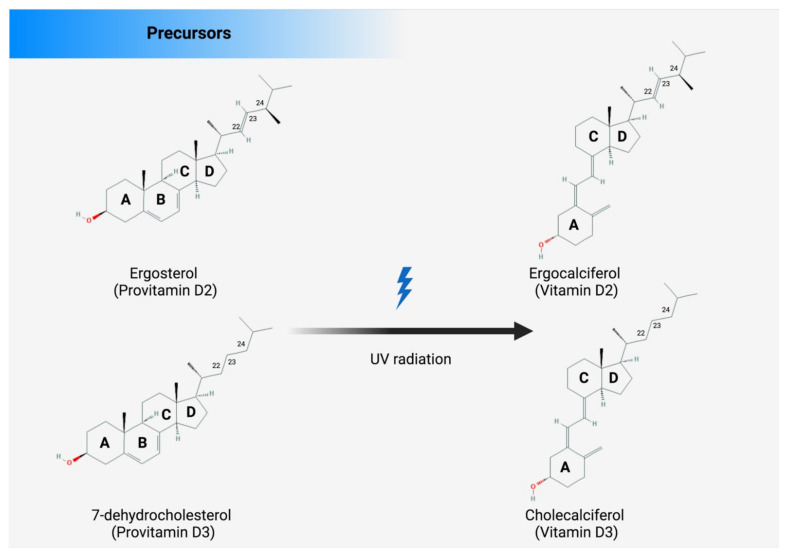
The structures of Vit D2 and Vit D3 and their precursors. The structural difference between Vit D2 and D3 is in the side chain. Compared with cholecalciferol, Vit D2 has a double bond between carbons 22 and 23 and a methyl group on carbon 24 of the broken ring.

**Figure 4 antioxidants-11-01090-f004:**
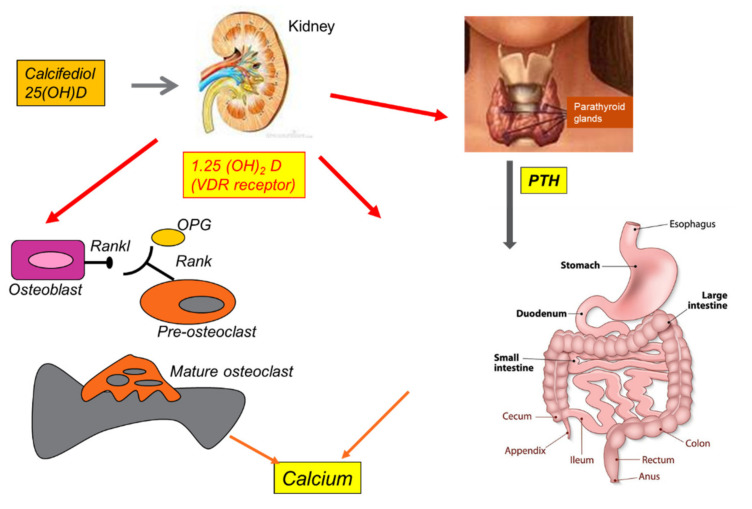
Summary of Vitamin D effects on human bone, thyroid, kidney, and calcium homeostasis.

**Figure 5 antioxidants-11-01090-f005:**
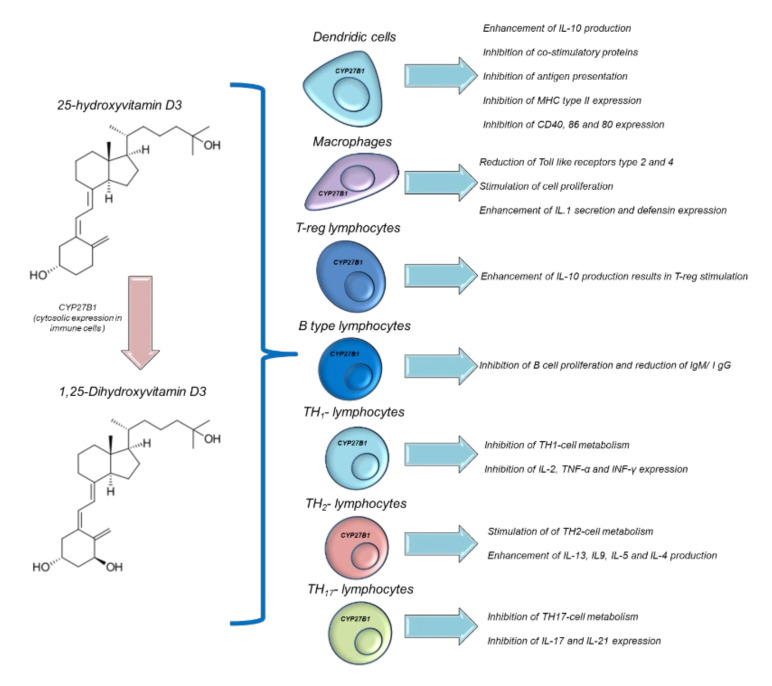
Immune modulation of Vitamin D in different immune cells.

**Table 1 antioxidants-11-01090-t001:** CYPs that are mainly involved in Vit D metabolism.

25-Hydroxylase Activity			
CYP Enzyme	Hydroxylate Function	Location/Expression	Note
CYP27A1	Does not 25-hydroxylate Vitamin D2	Liver, mitochondrial 25-hydroxylase, widely distributed throughout the body	Mouse mutation: increase in 25OHD blood level [21]. Human: inactivating mutations cause cerebrotendinous xanthomatosis, abnormal bile and cholesterol metabolism, but not rickets [34].
CYP2R1	25-hydroxylates both vitamin D2 and D3	Microsomal fraction of mouse liver [35], expression in liver and testes	CYP27A1 null mouse: increase CYP2R1 expression, hence increased blood levels of 25OHD. CYP2R1 null mouse: 50% decrease in 25OHD blood level [36]. Human: leu99pro mutation caused severe bone disease, rickets, and low 25OHD [37].
CYP3A4	Prefers 1αOHD to 25OHD as substrate	Liver and intestine	Is a major drug-metabolizing enzyme
CYP2J3	More 25-hydroxylase activity than CYP2J2	Rat liver	More 25-hydroxylase activity than its human homolog (CYP2J2)
CYP2J2	Less 25-hydroxylase activity than CYP2J3	Primarily in the heart	It is the human homolog of CYP2J3, functioning mainly as an arachidonic acid epooxygenase [38].
CYP2D25	Does not have substantial 25-hydroxylase activity [34]	Its human homolog was initially isolated from pig livers and kidneys [39]	-
CYP2C11	25-hydroxylase activity for vitamin D3 and D2 and 1OHD analogs	Expression in livers of male rats	Known for hydroxylation of testosterone [40].
1α-hydroxylase			
CYP27B1	The only enzyme with 25OHD 1α-hydroxylase ativity	Kidney (mainly), other tissues such as epithelial cells in the lungs, breast, skin, intestine, prostate; parathyroid gland), pancreatic islets, thyroid, testes, ovary, and placenta; macrophages, and T and B lymphocytes and dendritic cells (DCs); osteoblasts and chondrocytes [37]	Gene mutation leads to pseudovitamin D deficiency caused by inadequate 1,25(OH)_2_D production [41,42]. Parathyroid hormone, fibroblast growth factor 23, and 1,25(OH)_2_D play important roles in 1α-hydroxylase regulation: PTH stimulates CYP27B1. FGF23 and 1,25(OH)_2_D inhibit CYP27B1 Elevated calcium arrests PTH, leading to CYP27B1 suppression. High phosphate stimulates FGF23 and leads to CYP27B1 termination [43].
24-hydroxylase			
CYP24A1	The only enzyme with 24-hydroxylase ativity	Kidneys (mainly)	With both 24-hydroxylase and 23-hydroxylase activity, CYP24A1 mutation leads to removal of all 24-hydroxylated metabolites of vitamin D; therefore, it can cause defective mineralization of intramembranous (not endochondral) bone [44]. It is a marker of 1,25(OH)_2_D response in cells that express it. Main role of CYP24A1 is to regulate levels of 1,25(OH)_2_D in tissues. Moreover, a number of malignancies have elevated CYP24A1 expression [45]. In tumours, its inhibition can increase 1,25(OH)2D levels, leading to antiproliferative/prodifferentiating [46].
3-epimerase (3-epi)			
-	It produces the 3- epi form of 1,25(OH)_2_D in keratinocyte [47]	Keratinocyte, hepatocyte-derived cells, parathyroid cells, colon cancer cells, osteo-blasts, but not in the kidneys	The enzyme has not been purified and sequenced yet. It therefore remains unclear whether one gene product is involved.
20-hydroxylation			
CYP11A1	20-hydroxylation of vitamin D	Keratinocyte	The product, 20-hydroxylation, or its metabolite, 20,23(OH)2D, appear to have similar activity to 1,25(OH)2D, although not for all functions [23].

**Table 2 antioxidants-11-01090-t002:** Major and moderate drug interactions with vitamin D [68].

Drug/Susbance	Action	Clinical Action	Interaction
Erdafitinib	The mechanism appears to be related to the pharmacodynamic effects of fibroblast growth factor receptor (FGFR/aKlotho) inhibition by erdafitinib.	Minimizes risk of hyperphosphatemia; it is recommended to restrict phosphate intake to 600–800 mg daily	Major
Ergocalciferol and Vit D3 derivates	Additional effects result in toxicity, manifesting as hypercalcemia, hypercalciuria, and hyperphosphatemia.	In hypercalcemia, vitamin D and any calcium supplements should be immediately stopped	Major
Oxcarbazepin	It induces CYP450 inducers. It may decrease the pharmacologic effects of vitamin D analogs, inducing the hepatic conversion of vitamin D to inactive metabolites.	Patients who metabolize CYP450 poorly must be supplemented with double doses of Vit D when receiving oxicarbazepin.	Moderate
Magnesium salts and products	Possibly increases plasma hypermagnesemia, particularly in chronic renal dialysis patients, due to potentially additive pharmacologic effects. Chronic hypermagnesemia may have a role in the pathogenesis of adynamic bone disease in dialysis patients.	Patients on chronic renal dialysis treated with a vitamin D analog should avoid magnesium-containing products	Moderate
Indapamide and other thiazide diuretics	Thiazide diuretics inhibit the renal excretion of calcium and may also enhance the responsiveness of bone and renal tubule to parathyroid hormone.Thus, the concurrent use of large amounts of calcium or vitamin D can lead to excessively high plasma levels of calcium.	Serum calcium should be monitored if patients experience signs of hypercalcemia, such as dizziness, weakness, lethargy, headache, myalgia, anorexia, nausea, vomiting, and seizures	Moderate

**Table 3 antioxidants-11-01090-t003:** Characteristics of *VDR* PCR-restriction fragment length polymorphism (PCR-RFLP) fragments.

Restriction Sites	PCR Amplicon	Allelic Variants	Fragments Post-Digestion (bp)
*Apa*I	745 bp	A/A	745
		A/a	745 + 528 + 218
		a/a	528 + 218
*TaqI*	745 bp	T/T	745
		T/t	745 + 293 + 251 + 201
		t/t	293 + 251 + 201
*BmsI*	822 bp	B/B	822
		B/b	822 + 646 + 176
		b/b	646 + 176
*FokI*	267 bp	f/f	267
		F/f	267 + 197 + 70
		F/F	197 + 70

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
