# Peer review of "The Multiple Effects of Vitamin D against Chronic Diseases: From Reduction of Lipid Peroxidation to Updated Evidence from Clinical Studies"

_antioxidants, 2022, doi:10.3390/antiox11061090_

Round 1

Reviewer 1 Report

This is a really comprehensive review about the possible role of vitamin D, or deficiency of vitamin D, in the aetiology and or treatment of a wide range of diseases. The authors have correctly summarised the many conclusions from the multitude of publications, that although vitamin D status is often well associated with a disease process, the evidence for vitamin D deficiency being a causative factor or vitamin D therapy being the key  method of treatment is still inconclusive. This manuscript, however, has a number of minor sentence construction errors which need to be corrected. Some of these are listed below:

Line 62: “vitamin D is transported to the nucleus” should read: “1,25(OH)2D is transported to the nucleus”

Line 77: “effect of Vit D in on human health.” Should read “effect of Vit D on human health.”

Line 98: “Vit D can be synthesized” should read: “Vit D can be obtained”

Line 103: “from the sun through the formation of its precursor”. Should read: “ from the sun on its precursor.”

Line 104: “In plants and fungi…” is incorrect ergosterol is not present in most green plants so fungi are really the only source of ergosterol.

Line 256: “depending of the presence or the absence” should read: “depending on the presence or the absence”

Line 445: “e.i.” should probably be “i.e.”

Line 531: “One of the most intuitively neoplasm which could be related to any Vit D3 level or metabolism alteration is skin cancer.” This sentence is very difficult to follow. Perhaps it would be better written as: “One of the neoplasms which most intuitively could be related to vitamin D status or alteration in vitamin D metabolism, is skin cancer.”

There are many places in the text which refer to ‘Vit D serum level” This is incorrect. Both vitamin D and 25-hydroxyvitamin D are present in serum. The concentration that is important in determining vitamin D status is that of 25-hydroxyvitamin D. Hence the term “Vit D serum level” should read “25OHD serum level” This error is found at lines 553, 557, 560, 564, 570, 591, 598, 601, 627, 636, 745, 752, 755, 773.

Line 663: Contrary, several evidence indicates that…” would be better written as :Several pieces of evidence indicate that…”

Line 698: The word “Contary” is incorrect and does not seem relevant here. It would be better to simply say: “Patients with COVID-19 were found to have….”

Line 707: the word “interventistic” is incorrect. Perhaps it should read “interventional”

Author Response

Dear editor,

Thank you for the constructive comments to our manuscript. We have answered all the issues that were raised by the reviewers and improved the manuscript according to their suggestions. Please find below our reply. Track changes are included in the revised manuscript.

Reviewer 1

This is a really comprehensive review about the possible role of vitamin D, or deficiency of vitamin D, in the aetiology and or treatment of a wide range of diseases. The authors have correctly summarized the many conclusions from the multitude of publications, that although vitamin D status is often well associated with a disease process, the evidence for vitamin D deficiency being a causative factor or vitamin D therapy being the key  method of treatment is still inconclusive. This manuscript, however, has a number of minor sentence construction errors which need to be corrected. Some of these are listed below:

Line 62: “vitamin D is transported to the nucleus” should read: “1,25(OH)2D is transported to the nucleus”

Line 77: “effect of Vit D in on human health.” Should read “effect of Vit D on human health.”

Line 98: “Vit D can be synthesized” should read: “Vit D can be obtained”

Line 103: “from the sun through the formation of its precursor”. Should read: “ from the sun on its precursor.”

Line 104: “In plants and fungi…” is incorrect ergosterol is not present in most green plants so fungi are really the only source of ergosterol.

Line 256: “depending of the presence or the absence” should read: “depending on the presence or the absence”

Line 445: “e.i.” should probably be “i.e.”

Line 531: “One of the most intuitively neoplasm which could be related to any Vit D3 level or metabolism alteration is skin cancer.” This sentence is very difficult to follow. Perhaps it would be better written as: “One of the neoplasms which most intuitively could be related to vitamin D status or alteration in vitamin D metabolism, is skin cancer.

There are many places in the text which refer to ‘Vit D serum level” This is incorrect. Both vitamin D and 25-hydroxyvitamin D are present in serum. The concentration that is important in determining vitamin D status is that of 25-hydroxyvitamin D. Hence the term “Vit D serum level” should read “25OHD serum level” This error is found at lines 553, 557, 560, 564, 570, 591, 598, 601, 627, 636, 745, 752, 755, 773.

Line 663: Contrary, several evidence indicates that…” would be better written as :Several pieces of evidence indicate that…”

Line 698: The word “Contary” is incorrect and does not seem relevant here. It would be better to simply say: “Patients with COVID-19 were found to have….”

Line 707: the word “interventistic” is incorrect. Perhaps it should read “interventional”

We thank the reviewer for the proper suggestions. We have made the required modifications to the manuscript and changed the sentences as suggested (Changes are highlighted in yellow).

Reviewer 2 Report

Dear authors,

After the review process, I have several comments: the abstract should be expanded; you should mention how they realized the figures and include the copyright; you should expand section 5 and the role of vitamin D in neurodegenerative diseases; the role of vitamins in neurodegenerative disease is essential in maintaining homeostasis in humans in the case of increased oxidative stress; section 13 - Vitamin D and human microbiota should be expanded and add more new findings of bioavailability of this functional compound and the modulation of microbial and metabolomic patterns; you should check the paper for errors in the presentation of microbial strains name, for example, section 13.

Best regards!

Author Response

Dear editor,

Thank you for the constructive comments to our manuscript. We have answered all the issues that were raised by the reviewers and improved the manuscript according to their suggestions. Please find below our reply. Track changes are included in the revised manuscript.

Reviewer 2

Dear authors,

After the review process, I have several comments: the abstract should be expanded; you should mention how they realized the figures and include the copyright; you should expand section 5 and the role of vitamin D in neurodegenerative diseases; the role of vitamins in neurodegenerative disease is essential in maintaining homeostasis in humans in the case of increased oxidative stress; section 13 - Vitamin D and human microbiota should be expanded and add more new findings of bioavailability of this functional compound and the modulation of microbial and metabolomic patterns; you should check the paper for errors in the presentation of microbial strains name, for example, section 13.

We thank the reviewer for the comments. As suggested we expanded the abstract as well as section 5 (Vitamin D and neurodegenerative diseases) and section 13 (Vitamin D and gut microbiota). In addition, errors in the microbial strains were corrected. All the changes are highlighted in green.

Round 2

Reviewer 2 Report

No other comments.